# Identity-Based Proxy Signature with Message Recovery over NTRU Lattice

**DOI:** 10.3390/e25030454

**Published:** 2023-03-04

**Authors:** Faguo Wu, Bo Zhou, Xiao Zhang

**Affiliations:** 1Key Laboratory of Mathematics, Informatics and Behavioral Semantics (LMIB), Beihang University, Beijing 100191, China; 2Institute of Artificial Intelligence, Beihang University, Beijing 100191, China; 3Zhongguancun Laboratory, Beijing 100194, China; 4Bejing Advanced Innovation Center for Future Blockchain and Privacy Computing, Beihang University, Beijing 100191, China; 5School of Mathematical Sciences, Beihang University, Beijing 100191, China

**Keywords:** lattice-based cryptography, proxy signature, message recovery, post quantum resistant

## Abstract

Proxy signature is one of the important primitives of public-key cryptography and plays an essential role in delivering security services in modern communications. However, existing post quantum proxy signature schemes with larger signature sizes might not be fully practical for some resource-constrained devices (e.g., Internet of Things devices). A signature scheme with message recovery has the characteristic that part or all of the message is embedded in the signature, which can reduce the size of the signature. In this paper, we present a new identity-based proxy signature scheme over an NTRU lattice with message recovery (IB-PSSMR), which is more efficient than the other existing identity-based proxy signature schemes in terms of the size of the signature and the cost of energy. We prove that our scheme is secure under a Short Integer Solution (SIS) assumption that is as hard as approximating several worst-case lattice problems in the random oracle model. We also discussed some application scenarios of IB-PSSMR in blockchain and Internet of Things (IOT). This paper provides a new idea for the design of lattice signature schemes in low resource constrained environments.

## 1. Introduction

Proxy signature scheme is an emergency backup strategy of digital signatures, which can designate an agent to continue to perform signature verification in the absence of the signer. It was first proposed by Mambo, Usuda, and Okamoto et al. [1] in 1996. Subsequently, proxy signatures have been widely used in many scenarios, such as anonymous voting, electronic cash, mobile agents, etc. In the design of the construction scheme, most of the construction ideas are based on the difficult problems of traditional number theory, such as the difficult problems of (Elliptic Curve) discrete logarithms and factorization of large integers [2,3]. However, in the era of quantum computers, we need to find solutions based on other difficult problems, because these traditional schemes will be cracked by quantum algorithms in polynomial time [4]. Under this threat, many scholars began to study post quantum cryptography to prevent many important cryptosystems from failing directly after the advent of quantum computers. In the specific structure, there are mainly the following categories: lattice cryptography, multivariable cryptography, code-based cryptography, and Hash-based cryptography. Accordingly, some proxy signatures with post quantum security have been proposed, such as [5,6,7,8,9].

Lattice-based signature schemes have attracted many scholars’ attention, as their difficulty assumptions rely on some math problems that have been widely studied and come with uniquely strong security guarantees where lattice cryptosystems, on average (i.e., with randomly chosen keys), are as hard as the hardest problem of the underlying lattice problem [10]. Furthermore, In lattice cryptography, the operations involved in key generation, encryption, or signature usually involve only vector multiplication or modular addition over the integer ring, which makes the implementation of the scheme relatively simple. However, most lattice-based proxy signatures have large signature sizes, which makes lattice-based proxy signatures unsuitable in resource-constrained environments. Reducing the signature length is the most difficult problem in the practical application of lattice signatures, and how to solve and improve this problem is a critical question.

Traditional digital signature schemes usually need to bind messages and signatures to facilitate verifiers to verify them. This may incur additional bandwidth costs, especially when the message and signature sizes are relatively large. Scholars began to think about how to compress the size of messages and signatures as much as possible to reduce bandwidth consumption. The concept of message recovery was born in this case. Through message recovery, messages will be embedded in the signature. The sender sends the embedded signature to the receiver. After receiving the signature, the receiver can recover the original message from the signature and then perform signature verification. This construction method is very suitable for environments where signature size is required or bandwidth is limited [11,12]. In 1993, Nyberg and Ruppel modified the Digital Signature Algorithm (DSA) to support message recovery. It was the first signature scheme to support message recovery [13]. This has caused many scholars to pay attention to message recovery. Based on the lattice-based signature scheme of Lyubashevsky et al.  [14], Tian et al. [15] constructed a scheme supporting message recovery on the lattice, allowing them to have more advantages in communication bandwidth than Lyubashevsky et al., but Tian et al.’s scheme does not support proxy for signing rights. In 2017, Faguo Wu et al. [16] considered the problem of signature authority proxy and constructed the first lattice based proxy signature scheme using public key infrastructure. In addition, their scheme supports message recovery, and then has a good performance in communication overhead. In 2019, Xiuhua Lu et al. [17] considered identity-based settings and constructed a proxy signature with message recovery over lattices. However, Refs. [16,17] are based on inefficient lattice structures, and these schemes are trapped in large signature sizes. People naturally think about how to construct efficient schemes with lattices. As far as we know, the  NTRU lattice is the most efficient lattice. At present, it is still an open question whether the NTRU lattice can be used to construct a signature scheme with message recovery.

In terms of signature schemes designed based on quantum computing, Feng et al. [18] proposed a new quantum group signature scheme to enhance the non-repudiation of signatures. Lu et al. [19] proposed a verifiable arbitration quantum signature scheme based on controlled quantum teleportation, which can realize eavesdropping detection and identity authentication. Chen et al. [20] proposed a quantum multi-proxy blind signature based on cluster states to achieve blindness, non-repudiation and unforgeability. Feng et al. [21] studied an arbitrated quantum signature protocol based on boson sampling, which can resist forgery attack and denial attack. Feng et al. [22] proposed a quantum signature scheme for teleportation arbitration based on quantum walks, in which the entangled state is generated at the signature stage through quantum walks.

For the concrete application, Fang et al. [23] surveyed the application of proxy signatures in blockchain and investigated their usage in payment and integrity verification. In order to meet the challenges of data authentication and integrity in the Internet of Things environment, Verma et al. [24] proposed the first certificate-based proxy signature scheme without pairing. The proposed scheme is suitable for the Internet of Things in terms of computational cost. In the edge computing environment of the Internet of Things, resources are usually limited. Zhang et al. [25] proposed an ID-PRS scheme in the architecture of the Internet of Things, which also does not use pairing operations with high resource consumption, and supports non-interactive design. To address security and privacy issues in the Unmanned Aerial Vehicles (UAV) environment and mitigate various attacks, Verma and Singh et al. [26] proposed a short proxy signature scheme based on certificate setting, which has advantages in signature length and computational efficiency.

In this paper, inspired by the lattice-based signature schemes [15,16,27,28], we first propose an identity-based proxy signature with message recovery over the NTRU lattice. In the random oracle model, our scheme can achieve delegation information and signature existential unforgeability under adaptive chosen warrant and identity attacks. Since our signature scheme adopts message recovery technology, compared with some existing proxy signature schemes, our scheme has better performance in communication overhead and signature size. Finally, when we consider the actual application [29], we find that this scheme performs well in terms of energy consumption, which means that our scheme is very suitable for resource constrained and low bandwidth environments. Due to the hardness assumption of SIS over the NTRU lattice, we formally constructed a lattice-based message recovery proxy signature scheme that can provide post quantum security in the quantum era.

The rest of the article is arranged as follows. In Section 2, we provide necessary preliminaries of our scheme. In Section 3, we give a detailed description of the syntax model and security model of our identity-based proxy signature with message recovery. In Section 4, we formally show how we construct the basic message recovery proxy signature. In Section 5, we present the formal security analysis of our scheme. In Section 6, we introduce detailed comparisons between our scheme and some existing proxy schemes. In Section 7, we discuss some application scenarios of our proposed IB-PSSMR scheme. Finally, we conclude our paper in Section 8.

## 2. Preliminary Knowledge

### 2.1. Notations

In this article, we agree that these tokens represent the following specific meanings:   

•      ${∥v∥}_{p}$ denotes the $l_{p}$ norm of *v*.

•      $M^{n\times (k_{1}+k_{2})}=M_{1}^{n\times k_{1}}\|M_{2}^{n\times k_{1}}$ denotes the concatenation of Matrices $M_{1},M_{2}$.

•      ∣x∣ indicates the length of *x* under binary representation.

•      $|x{|}^{l_{1}}$ denotes the first left $l_{1}$ bits of *x*.

•      $|x{|}_{l_{2}}$ denotes the first right $l_{2}$ bits of *x*.

•      x‖y denotes string concatenation. It means append string *y* at the behind of string *x*

### 2.2. NTRU Lattice

Let $R_{q}$ be the ring $Z_{q}\left[x\right]/\left(x^{N}+1\right)$, and *f*,*g* be the polynomials in $R_{q}$. Let *h* be the polynomial convolution of $f^{-1}$ and *g*. In other words,
(1)$$h=f^{-1}g\;\mathrm{mod}\;\left(X^{N}+1\right)$$
where $f=\sum_{i=0}^{N-1}f_{i}x^{i}$ and $g=\sum_{i=0}^{N-1}g_{i}x^{i}$. The NTRU lattice associated with *h* and *q* is
(2)$${⋀}_{h,q}=\left\{\left(u,v\right):u+v*h\;\mathrm{mod}\;q=0\right\}$$

${⋀}_{h,q}$ is a full rank lattice in $Z^{2N}$ generated by the rows of
(3)$$A_{h,q}=\left(\begin{array}{cc} A_{N}\left(h\right) & I_{N}\\ qI_{N} & O_{N} \end{array}\right)$$
where $A_{N}\left(h\right)$ is an anticirculant matrix whose *i*th row consists of the coefficients of the polynomial $hx^{i}$ mod $\left(X^{N}+1\right)$. Additionally $I_{N}$ is the N×N unit matrix, $O_{N}$ is the N×N null matrix. We emphasize that NTRU lattices have some excellent properties: their Gram–Schmidt norm can be small and they can be computed quickly.

**Definition** **1.**
*Given integers q,m,n and a matrix $A\in Z_{q}^{n\times m}$, the ${}^{\prime }q-ary^{\prime }$ lattices are defined as follows*

$$\begin{array}{c} \Lambda_{q}\left(A\right)=\left\{x\in Z^{m}:x=A^{T}s\;\mathrm{mod}\;q,\;\mathrm{for}\;\mathrm{some}\;s\in Z_{q}^{n}\right\}\\ \Lambda_{q}^{⊥}\left(A\right)=\left\{x\in Z^{m}:A^{T}x=0\;\mathrm{mod}\;q\right\} \end{array}$$



$\Lambda_{q}\left(A\right)$ and $\Lambda_{q}^{⊥}\left(A\right)$ are dual to each other.

### 2.3. Gaussian on Lattice

In this section, we introduce an algorithm to sample the discrete Gaussian distribution, and the output result is a vector obeying the discrete Gaussian distribution. As shown in Algorithm 1.
**Algorithm 1** GaussianSampler**Input:** Lattice Λ basis *B*, standard deviation σ, center $c\in Z^{N}$**Output:** Vector *v* sampled in $D_{\Lambda ,\sigma ,c}$1:$v_{n}\leftarrow 0$2:$c_{n}\leftarrow c$3:**for** 
i=n,n−1,⋯,1 
**do**4:    $c_{i}^{{}^{\prime }}\leftarrow \langle c_{i},\tilde{b_{i}}\rangle /{\|\tilde{b_{i}}\|}^{2}$5:    $\sigma_{i}^{{}^{\prime }}\leftarrow \left\|\tilde{b_{i}}\right\|$6:    $z_{i}\leftarrow SampleZ\left(c_{i}^{{}^{\prime }},\sigma_{i}^{{}^{\prime }}\right)$7:    $c_{i-1}\leftarrow c_{i}-z_{i}b_{i}$8:    $v_{i-1}\leftarrow v_{i}-z_{i}b_{i}$9:**end for**10:**return** 
$v_{0}$

The subalgorithm SampleZ samples a 1-dimensional Gaussian $D_{Z^{N},\sigma ,c}$. There are various techniques for 1-dimensional discrete Gaussian sampling, such as the inverse method [30], the Knuth–Yao algorithm [31], rejection sampling [32] and discrete ziggurat algorithms [33].

According to Lyubashevsky’s discussion on Lattice trapdoor [28] construction, consider the discrete Gaussian distribution in dimension *m* and let its standard deviation be σ, he proposed some important properties of Discrete Gaussian distribution. We refer it as Lemma 1.

**Lemma** **1.**
*∀σ>0 and m∈Z*

*(1) $Pr\left[x\in D_{\sigma }^{1}:|x|>12\sigma \right]<2^{-100}$;*

*(2) $Pr[x\in D_{\sigma }^{m}:\|x\|>2\sigma \sqrt{m}]<2^{-m}$;*

*(3) For any $v\in Z^{m}$ and any positive real α, if $\sigma =\omega (\|v\|\sqrt{\log m})$, then we have the following probability relation.*

(4)
$$Pr\left[x\in D_{\sigma }^{m}:D_{\sigma }^{m}\left(x\right)/D_{\sigma ,v}^{m}=o\left(1\right)\right]=1-2^{\omega \log m}$$

*Additionally ω(.) is the non-asymptotic tight lower bound. More specifically, for a given quantity relationship, If σ=α‖v‖, we can obtain the following inequality relation.*

(5)
$$Pr\left[x\in D_{\sigma }^{m}:D_{\sigma }^{m}\left(x\right)/D_{\sigma ,v}^{m}<e^{12/\alpha +1/(2\alpha^{2})}\right]>1-2^{-100}$$



### 2.4. Rejection Sampling Technique

The Rejection Sampling Technique [10] is mainly used to eliminate the relationship between the signing key and output signature. The algorithm is described below.

If the signer follows the steps in Algorithm 2, then the distribution of the outputted signatures is $min(\frac{D_{\sigma }^{m}\left(z\right)}{MD_{\mathrm{Sc},\sigma }\left(z\right)},1)$ and the expected number of times that this process will output a signature is *M*.
**Algorithm 2** Rejection sampling technique**Input:** Message *u*, a matrix *A* randomly sampled from $Z_{q}^{m\times n}$, S(signature key) sampled from ${\left\{-d,\cdots ,0,\cdots ,d\right\}}^{m\times k}$, $H:{\left\{0,1\right\}}^{*}{\to \{v:v\in \left\{-1,0,1\right\}}^{k},\|v\|<\kappa \}$, where $d\ll q^{n/m}$, k∈Z and ≪m, κ is constant and 2κ·kκ≥2100. Then there exists a constant M=O(1).**Output:** Vector z and c1: Obtain y randomly from $D_{\sigma }^{m}$2: c=H(Ay,u)3: z=Sc+y **return** (z,c) with probability $min(\frac{D_{\sigma }^{m}\left(z\right)}{MD_{\mathrm{Sc},\sigma }\left(z\right)},1)$

### 2.5. Hardness Assumption

We assume the SIS problem is hard in the NTRU lattice, and referring to [34], when we choose *f* and *g* in key generation properly, the distribution of $h=f^{-1}g$ and uniform distribution of $R^{*}$ are statistically close to each other, which means they are indistinguishable. Here we recall the definition of the SIS problem.

**Definition** **2.**
*(Small Integer Solution problem (SIS)) Let n and q be integers, where n stands for the security parameter. Typically q is a polynomial of n. Let β>0. Given a uniformly random matrix $A\in Z_{q}^{n\times m}$ where m also satisfies m=poly(n), the goal is to find a non-zero vector $e\in Z^{m}$, such that Ae=0modq and ‖e‖<β.*


**Definition** **3.**
*Given f,g,h in NTRU’s key pair generation, n,q,β is defined the same as in Definition 2. The SIS problem over NTRU lattice is to find a non-zero vector $\left(z_{1},z_{2}\right)$, such that it satisfies $A_{h,q}\left(z_{1},z_{2}\right)=0$ mod q and $\|(z_{1},z_{2})\|<\beta$.*


Assume that $\left(s_{1},s_{2}\right)$ is any of the vectors in the $A_{h,q}$, the γ−SVP problem on the $A_{h,q}$ is to find the vector $\left(z_{1},z_{2}\right)$ satisfy $\|\left(z_{1},z_{2}\right)\left\|\leq \gamma \right\|\left(s_{1},s_{2}\right)\|,$ that is, $\|(z_{1},z_{2})\|\leq \gamma \theta$. Among which θ is the shortest length of the vector in lattice $A_{h,q}$. Therefore, when γ=β/θ, solving SIS over the NTRU lattice is as hard as solving the shortest vector problem in the NTRU lattice. Hence, we claim that our proposed scheme also relies on the hardness of γ−SVP. Note that the γ−SVP problem is NP-hard when the approximate factor $\gamma <1+1/n^{\varepsilon }$ [35].

### 2.6. Message Recovery

Message recovery is a function extension of the signature scheme, allowing all or part of the messages to be embedded in the signature. The key generation, signature, verification algorithms, and message recovery process are shown in the Figure 1.

Gen, Sign, and Ver are the Key generation algorithm, signature and verification algorithm, SK is the secret key and PK is the public key. Message *u* to be signed is divided into two parts $u=u_{1}\|u_{2}$. $u_{1}$ is the recoverable part that is embedded in the signature and can be recovered from the signature during the verification process, and the non-recoverable part $u_{2}$ can be sent or stored with the signature.

## 3. Syntax and Security Model for Identity-Based Proxy Signature Scheme with Message Recovery

In this section, we will first give the syntax model, i.e, we describe the participants in our scheme, and the algorithms in our scheme. Then, we introduce the security model of our lattice-based proxy signature scheme with message recovery(IB-PSSMR).

### 3.1. Syntax

**Definition** **4.**
*There are four types of participants in our identity-based proxy signature with message recovery over the NTRU lattice:*

*Original signer with ${\mathrm{ID}}_{o}$;*

*Proxy signer ${\mathrm{ID}}_{p}$;*

*Verifier;*

*Key generation center (KGC) in the system.*


*Our scheme consists of six probabilistic polynomial-time (PPT) algorithms (*
*
**Setup**
*
*, *
*
**KeyExtract**
*
*, *
*
**DelGen**
*
*, *
*
**DelVer**
*
*, *
*
**Psign**
*
*, and *
*
**Pver**
*
*), and their roles are as follows:*
***Setup****: The algorithm* ***Setup*** *takes a security parameters N as input, and then it outputs the system’s public parameters par, KGC’s public and secret key (mpk,msk), that is (par,(msk,mpk))←Setup(n).****KeyExtract****: The algorithm* ***KeyExtract*** *takes the system’s public parameters par, KGC’s secret key msk and public key mpk, user’s identity (i.e., user’s public key pk) ${\mathrm{ID}}_{u}$ as input, and then it outputs the user ${\mathrm{ID}}_{u}$’s secret key $sk_{ID}$, that is, $sk_{ID}\leftarrow \mathrm{KeyExtract}\left(par,msk,{\mathrm{ID}}_{u}\right)$.*
*
**DelGen**
*
*: The algorithm*
*
**DelGen**
*
*’s input consists of the system’s public parameters par, KGC’s public key mpk, a warrant W where $W=(pk_{ID_{o}},pk_{ID_{p}},T)$, T is valid time period of W, original signer’s secret and public key $\left(sk_{ID_{o}},pk_{ID_{o}}\right)$, original signer computes the delegation, it outputs the delegation information $d_{g}$, that is, $\left\{d_{g}\right\}\leftarrow \mathrm{DelGen}\left(par,W,mpk,sk_{ID_{o}},pk_{ID_{o}}\right)$.*

*
**DelVer**
*
*: On input the system’s public parameters par, KGC’s public key mpk, original signer’s public key $pk_{ID_{o}}$, warrant W and its delegation $d_{g}$, he verifies the legality of delegation information $d_{g}$, If delegation $d_{g}$ satisfied, the output is 1, and the delegation is accepted; otherwise, the output is 0, and the delegation is rejected, that is, $\left\{0,1\right\}\leftarrow \mathrm{DelVer}\left(par,W,d_{g},mpk,pk_{ID_{o}},pk_{ID_{p}}\right)$.*

*
**Psign**
*
*: Given the system’s public parameters par, KGC’s public key mpk, original signer’s public key $pk_{ID_{o}}$, proxy signer’s secret and public key $\left(sk_{ID_{p}},pk_{ID_{p}}\right)$, delegation key $\left(sk_{d},pk_{d}\right)$, warrant W and delegation information $d_{g}$, and the message m to be signed, the algorithm*
*
**Psign**
*
*outputs the identity-based proxy signature(IB-PS) on behalf of the original signer, that is, $sig\leftarrow \mathrm{Psign}(par,m,W,mpk,pk_{ID_{o}},sk_{ID_{p}},pk_{ID_{p}},sk_{d},pk_{d})$.*

*
**Pver**
*
*: For a verifier in our IB-PSSMR system, he first recovers the message m embedded in the signature sig. Then, the algorithm*
*
**Pver**
*
*takes the public key $pk_{ID_{o}}$ of the original signer, the public key $pk_{ID_{p}}$ of the proxy signer, and the public delegation key $pk_{d}$ as input. if the proxy signature is valid, output 1, or output 0 if it is invalid, that is $\left\{m,\left\{0,1\right\}\right\}\leftarrow \mathrm{Pver}\left(par,sig,pk_{ID_{o}},pk_{ID_{p}}\right)$.*



**Definition** **5.**
*Given security parameters n, to make our scheme IB-PSSMR work correctly, the six PPT algorithms should meet the following rules*


(par,(msk,mpk))←Setup(n)



sk←KeyExtract(par,msk,ID)



$\left\{sk_{d},pk_{d},d_{g}\right\}\leftarrow \mathrm{DelGen}\left(par,W,mpk,sk_{ID_{o}},pk_{ID_{o}}\right)$



$\left\{0,1\right\}\leftarrow \mathrm{DelVer}\left(par,W,d_{g},mpk,sk_{d},pk_{d},pk_{ID_{o}},pk_{ID_{p}}\right)$



$sig\leftarrow \mathrm{Psign}(par,m,W,mpk,pk_{ID_{o}},sk_{ID_{p}},pk_{ID_{p}})$



$\left\{m,\left\{0,1\right\}\right\}\leftarrow \mathrm{Pver}\left(par,sig,pk_{ID_{o}},pk_{ID_{p}}\right)$


*the above-mentioned algorithms hold with overwhelming probability.*


### 3.2. Security Model for IB-PSSMR

For the security issue of identity-based proxy signature scheme with message recovery (IB-PSSMR) over NTRU lattice, there are two things we should concern about. First, the delegation is the proxy signer’s signature on the message *m*, which is made on behalf of the original signer. Second, the warrant is a kind of timestamp restriction of message and contains the valid period of time. Considering this, Unforgeability, Verifiability, Strong identifiability, Strong undeniability, and Key dependence are naturally satisfied. Therefore, the security model of this IB-PSSMR over NTRU lattice is existential unforgeable under adaptive chosen-message attacks. We define the security model of our IB-PSSMR by a game, or an experiment, run between a challenger C and an adversary A(forger).

In regard to the unforgeability of our IB-PSSMR over NTRU lattice, we should take two types of adversary into consideration:

Type(i): Adversary A can obtain access to the original signer’s public key $pk_{ID_{o}}$, proxy signer’s public key $pk_{ID_{p}}$,original signer’s secret key $sk_{ID_{o}}$.

Type(ii): Adversary A can not obtain access to the original signer’s secret key $sk_{ID_{o}}$, proxy signer’s secret key $sk_{ID_{p}}$.

It is evident that the adversary in Type(i) is more powerful than the adversary in Type(ii), thus we will only consider the Type(i) adversary.

The security game of the IB-PSSMR is defined by the interactions between a challenger C and an adversary A. Additionally, the interactions consist of the following phases:Initial Phase: the challenger C runs the Setup(n) algorithm to generate the system public parameters par and then C sends them to the adversary A.Query Phase: in the Query Phase, the adversary A can adaptively issue some query (also known as query the oracles). The number of queries is polynomial bounded.KeyExtract-query: given an ID, the adversary A can issue a query to obtain the corresponding secret key. The challenger C runs the algorithm $sk_{ID}\leftarrow \mathrm{DelGen}\left(par,W,mpk,sk_{ID_{o}},pk_{ID_{o}}\right)$, and returns A with $sk_{ID}$.DelGen-query: for some interested delegation information $d_{g}$, the adversary A issues query with two secret key corresponding to the identity $ID_{o}$ and $ID_{p}$ as input. Once upon receiving the query, the challenger C runs $d_{g}\leftarrow \mathrm{DelGen}\left(par,W,mpk,sk_{ID_{o}},pk_{ID_{o}}\right)$. Additionally, C returns $d_{g}$ to A.Psign-query: if A is interested in the proxy signature of message *m* under $ID_{p}$, he issues such a query to the challenger. C runs the algorithm $\mathrm{sig}\leftarrow \mathrm{Psign}(par,m,W,mpk,pk_{ID_{o}},sk_{ID_{p}},pk_{ID_{p}})$, and delivers sig to A.Forgery Phase: through the query phase above, the adversary A tries to forge a proxy signature to win the game. Given a message *m* and an identity $ID_{p}$ as the proxy signer, A needs to generate a valid sig to make it pass the verification. The following conditions should naturally be satisfied:(a)$\mathrm{Pver}(par,pk_{ID_{o}},pk_{ID_{p}})=1$.(b)In the Psign-query phase, *m* has never been signed.(c)In the KeyExtract-query phase, the secret key of $ID_{p}$ has not been queried.

**Definition** **6.**
*If the advantage of any PPT adversary A wins the security game above is negligible, then the Identity-based proxy signature with message recovery(IB-PSSMR) over NTRU lattice is regarded as existential unforgeable.*


## 4. Our Identity-Based Proxy Signature Scheme with Message Recovery

The identity-based proxy signature scheme with message recovery (IB-PSSMR) over NTRU lattice we proposed is discussed in this section. There are four participants in our scheme:A trusted third party KGC,An original signer with $ID_{o}$,A proxy signer with $ID_{p}$,A verifier.

Additionally, our scheme IB-PSSMR over NTRU lattice consists of six probabilistic polynomial time algorithms (Setup, KeyGen, DelGen, DelVer, Psign, and Pver), where:Setup: the Setup algorithm run by KGC. It takes a system security parameter λ as the algorithms’ input. Assume q≥3, λ,N be positive integers. The Setup algorithm will do the following steps:Choose hash functions $H_{1}:{\left\{0,1\right\}}^{*}\to Z_{q}^{N}$, $H_{2}:Z_{q}^{n}\to {\left\{0,1\right\}}^{l_{1}+l_{2}}$, $H_{3}:{\left\{0,1\right\}}^{*}\to {\left\{0,1\right\}}^{N\times N}$, $l_{1}$, $l_{2}$∈N, $H_{2}$, $H_{3}$ are seen as a random oracle.Select two encoding functions $F_{1}:{\left\{0,1\right\}}^{l_{2}}\to {\left\{0,1\right\}}^{l_{1}}$, $F_{2}:{\left\{0,1\right\}}^{l_{1}}\to {\left\{0,1\right\}}^{l_{2}}$.KGC starts the algorithm MasterKeygen to output the system’s master key (msk,mpk), which is described in Algorithm 3.Finally, KGC publishes $par=(N,q,H_{1},H_{2},H_{3})$ as public parameters of our IB-PSSMR system.KeyExtract: KGC takes the public parameters par and system’s master secret key msk as the algorithm’s input, then KGC works as follows:The system’s participants original signer and proxy signer request their secret key from KGC, and offer their identity $ID_{o}$ and $ID_{p}$, respectively.KGC first checks whether these identities exist in the identity list IDLIST. If so, KeyExtract request can be terminated, otherwise, KGC runs GaussianSampler$\left(c,\sigma ,\left(H_{1}\left(ID_{o}\right),0\right)\right)$ to obtain $ID_{o}$’s secret key ${\mathrm{sk}}_{o}=\left(s_{1},s_{2}\right)$ and runs Gaussian$\mathrm{Sampler}(c,\sigma ,\left(H_{1}\left(ID_{p}\right),0\right))$ to obtain $ID_{p}$’s secret key ${\mathrm{sk}}_{p}=\left(s_{3},s_{4}\right)$, where $s_{1}+s_{2}h=H_{1}\left(ID_{o}\right)$ and $s_{3}+s_{4}h=H_{1}\left(ID_{p}\right)$.KGC sends $sk_{p}$ to the proxy signer and $sk_{o}$ to the original signer by a a secure authenticated channel.DelGen: original signer generates the delegation on warrant *W* where $W=(pk_{ID_{o}},pk_{ID_{p}},T)$, *T* is the valid time period of *W*, and delegation information $d_{g}$ on *W* is described as Algorithm 4.DelVer: when the proxy signer receives the warrant *W* and its delegation $d_{g}=\left(z_{1},z_{2}\right)$, he first checks if $\|\left(z_{1},z_{2}\right)\|\leq 2\sigma \sqrt{2N}$ and $H_{2}\left(hy_{2}+y_{1}-H_{1}\left(ID_{O}\right)*W,W\right)$ both are true. If the conditions hold, then proxy signer $ID_{p}$ can take the warrant as his lawful authority from the original signer; otherwise, he should reject it.Psign: after confirming the legitimacy of the signer, given a message *u*, the proxy signer with $ID_{p}$ can generate a proxy signature for it by Algorithm 4.Pver: given the public parameters par, for a a user in the system who wants to verify the legitimacy of the proxy signature, he performs the steps described in Algorithm 5.

**Theorem** **1.**
*The IB-PSSMR we proposed satisfies correctness.*


**Proof.** From the Algorithms 3–5’s detailed construction, we can easily have the following equations.
$$\begin{array}{cc} \; & H_{2}\left(hz_{i+1}+z_{i}-H_{1}\left(ID\right)H_{3}\left(r,u_{2}\right)\right)\\ \; & =H_{2}\left(h\left(s_{i+1}C+y_{2}\right)+\left(s_{i}C+y_{1}\right)-\left(s_{i+1}h+s_{i}\right)C\right)\\ \; & =H_{2}\left(y_{1}+y_{2}h\right)\\ \; & =\alpha \end{array}$$
the distribution of $\left(z_{i+1},z_{i}\right)$ and the distribution $D_{Z^{N},s}$ are statistically close to each other. By the Lemma 1, $\|z_{i}\|\leq 2\sigma \sqrt{N}$ with probability at least $1-2^{-m}$, that is, $\|\left(z_{i+1},z_{i}\right)\|\leq 2\sigma \sqrt{2N}$ satisfied with overwhelming probability. Furthermore, $u_{1}^{\prime }=F_{1}\left(u_{1}\right)\|\left(F_{2}\left(F_{1}\left(u_{1}\right)\right)⨁u_{1}\right)$, we can recover $u_{1}=|u_{1}^{\prime }{|}_{l_{2}}⨁F_{2}\left(\right|u_{1}^{\prime }{|}^{l_{1}})$ with $F_{1}\left(u_{1}\right)={\left|u_{1}^{\prime }\right|}^{l_{1}}$ hold.    □

**Algorithm 3** Master Keygen
**Input:** Security parameter *N*, prime *q*, σ**Output:** KGC’s public key mpk and secret key msk.
1:Start Sample $f,g\in D_{Z^{N},\sigma }$.2:**if** $\|f\|>\sigma \sqrt{N}$ or $\|g\|>\sigma \sqrt{N}$ or *f*mod*q* $\notin R_{q}^{*}$ or *g*mod*q*$\notin R_{q}^{*}$ **then**3:    Restart4:
**end if**
5:**if** max($\left\|\left(g,-f\right)\|,\right\|\left(\frac{g\bar{f}}{f\bar{f}+g\bar{g}},\frac{g\bar{g}}{f\bar{f}+g\bar{g}}\right)\left\|\right)>1.17\sqrt{g}$ **then**6:    Restart7:
**end if**
8:$R_{f}=\mathrm{resultant}\left(f,X^{N}+1\right)$ and $R_{g}=\mathrm{resultant}\left(g,X^{N}+1\right)$, respectively. The resultant of *f* can be straightforwardly calculated as $\prod_{i=1}^{N-1}f\left(X^{i}\right)$ (mod Φ(N)) where Φ(N) is the cyclotomic polynomial $\Phi \left(N\right)=1+X+X^{2}+\cdots +X^{N-1}$. The details of the resultant operation can refer to [36]9:Compute $\rho_{f},\rho_{f}$ satisfy $\rho_{f}f+k_{f}\left(X^{N}+1\right)=R_{f}$, $\rho_{g}f+k_{g}\left(X^{N}+1\right)=R_{g}$ by the Extended Euclidean Algorithm where $k_{f}$ and $k_{g}$ are integers.10:**if** 
$(R_{f},R_{g})\neq 1$
 **then**11:    Restart12:
**end if**
13:Use the Extended Euclidean Algorithm to find α and β satisfy $\alpha R_{f}+\beta R_{g}=1$, that is, we have $\left(\alpha \rho_{f}\right)f+\left(\beta \rho_{g}\right)g=1+k\left(x^{N}+1\right).$14:Let $F=q\beta \rho_{g}$, $G=-q\alpha \rho_{f}$, then f∗G−g∗F=q (mod $X^{N}+1$)15:**return** The KGC’s master public key $mpk=h=f^{-1}g$, KGC’s master secret key $msk=B=\left(\begin{array}{cc} A_{g} & -A_{f}\\ A_{G} & -A_{F} \end{array}\right)$, where $A_{g}$, $-A_{f}$, $A_{G}$ and $-A_{F}$ are anti-circulant matrices, and their *i*th row consists of the coefficients of the polynomial $gx^{i}\;\mathrm{mod}\;\left(X^{N}+1\right)$, $fx^{i}\;\mathrm{mod}\;\left(X^{N}+1\right)$, $Gx^{i}\;\mathrm{mod}\;\left(X^{N}+1\right)$ and $Fx^{i}\;\mathrm{mod}\;\left(X^{N}+1\right)$, respectively.


**Algorithm 4** Message recovery
**Input:** Private key $sk=(s_{i},s_{i+1})$, message *u***Output:** Message recovery signature $\left(z_{i},z_{i+1}\right)$
1:Choose $y_{1},y_{2}\in D_{Z^{N},\sigma }$2:Divide the message *u* into two parts $u=u_{1}∥u_{2}$ and make $|u_{1}|=l_{2}$, if $|u|<l_{2}$ then let $u_{2}=⊥$.3:Compute $\alpha =H_{2}\left(y_{1}+y_{2}h\right).$4:Compute $u_{1}^{{}^{\prime }}=F_{1}\left(u_{1}\right)\|\left(F_{2}\left(F_{1}\left(u_{1}\right)\right)⨁u_{1}\right).$5:Compute $r=\alpha ⨁u_{1}^{{}^{\prime }}.$6:Compute $C=H_{3}\left(r,u_{2}\right)$7:Compute $z_{i}=s_{i}C+y_{1},z_{i+1}=s_{i+1}C+y_{2}.$8:**if** Nothing is outputted **then**9:    Restart10:
**end if**
11:**return**$\left(u_{2},z_{i},z_{i+1}\right)$ on message *m* with probability $min(\frac{D_{Z^{N},\sigma }}{MD_{Z^{N},\sigma ,sku}},1)$, where M=O(1).


**Algorithm 5** Pver
**Input:** 

$r,z_{i},z_{i+1},u_{2}$

**Output:** 0 or 1
1:Compute $\alpha =H_{2}\left(hz_{i+1}+z_{i}-H_{1}\left(ID\right)H_{3}\left(r,u_{2}\right)\right)$2:Compute $u_{1}^{\prime }=r⨁\alpha$3:

$u_{1}=|u_{1}^{\prime }{|}_{l_{2}}⨁F_{2}\left(\right|u_{1}^{\prime }{|}^{l_{1}})$

4:Compute $u=u_{1}\|u_{2}$5:**if**$\|\left(z_{i},z_{i+1}\right)\|\leq 2\sigma \sqrt{2N}$, $F_{1}\left(u_{1}\right)={\left|u_{1}^{\prime }\right|}^{l_{1}}$**then**6:    Return 17:
**else**
8:    Return 09:
**end if**



## 5. Security Analysis

In this section, we give a formal proof to show that our proxy signature is unforgeable. If not, the adversary can break the hardness problem SIS in the NTRU lattice.

**Theorem** **2.**
*The proposed IB-PSSMR over NTRU lattice is existential unforgeable against adaptive chosen message and address attacks in the random oracle model under the hardness assumption of SIS problem over NTRU lattice.*


**Proof.** We prove the security of our scheme by contradiction. Suppose that if there is a PPT adversary A who can break our IB-PSSMR over NTRU lattice with non-negligible probability, we show that the adversary A can then solve the SIS problem over NTRU lattice.The security game can be described between a challenger C and an adversary A. We simulate the interaction between challenger C and adversary A as follows:Initial Taking λ as the security parameter, the algorithm C first randomly picks a matrix *h*, three secure hash functions $H_{1}:{\left\{0,1\right\}}^{*}\to Z_{q}^{N}$, $H_{2}:Z_{q}^{n}\to {\left\{0,1\right\}}^{l_{1}+l_{2}}$, $H_{3}:{\left\{0,1\right\}}^{*}\to {\left\{0,1\right\}}^{N\times N}$ and two encoding functions $F_{1}:{\left\{0,1\right\}}^{l_{2}}\to {\left\{0,1\right\}}^{l_{1}}$, $F_{2}:{\left\{0,1\right\}}^{l_{1}}\to {\left\{0,1\right\}}^{l_{2}}$ then sends the public parameters par={*h*, $H_{1}$, $H_{2}$, $H_{3}$, $F_{1}$, $F_{2}$} to the adversary A.Queries: The adversary A issues the following queries adaptively.
$H_{1}$-query: to make use of the $H_{1}$ oracle response, the challenger C builds a list $L_{0}$ to store the query response information. It is initialized as an empty set. Given the adversary’s $H_{1}$ query with ${\mathrm{ID}}_{i}$, C first check if it is in the list $L_{0}$. If there is a value corresponding to $H_{1}\left({\mathrm{ID}}_{i}\right)$, then return it to the adversary. Otherwise the challenger randomly chooses $H_{1}\left({\mathrm{ID}}_{i}\right)\in Z_{q}^{N}$, then updates the $H_{1}$ list $L_{0}$ as $L_{0}=\left(L_{0},\left\{{\mathrm{ID}}_{i},H_{1}\left({\mathrm{ID}}_{i}\right)\right\}\right)$, and finally outputs $H_{1}\left({\mathrm{ID}}_{i}\right)$ as the response.$H_{2}$-query: the challenger C maintains the $H_{2}$ list which is a list of tuples $L_{1}=\left(\alpha_{i},y_{i1}+y_{i2}h\right)$, and the initial value is null, when the adversary A issues a $H_{2}$ query on a vector $y_{i1}+y_{i2}h\in Z_{q}^{N}$, the challenger C looks it up in the $H_{2}$ list, if the challenger C finds a matched tuple $\left(\alpha_{i},y_{i1}+y_{i2}h\right)$, he returns $\alpha_{i}$ to adversary A as the query response. If not, C randomly selects string $\alpha_{i}\in {\left\{0,1\right\}}^{l_{1}+l_{2}}$, then updates the $H_{2}$ list $L_{1}$ as $L_{1}=\left(L_{1},\left\{\alpha_{i},y_{i1}+y_{i2}h\right\}\right)$, and finally outputs $\alpha_{i}$ as the response.$F_{1}$-query: the challenger C maintains a $F_{1}$ list $L_{2}=\left(u_{i1},F_{1}\left(u_{i1}\right)\right)$, and set it empty in the beginning. When there is a $F_{1}$ query for $u_{i1}$ from the adversary A, the challenger C first checks if it is in the $L_{2}$ list. If there is a corresponding pair $\left(u_{i1},F_{1}\left(u_{i1}\right)\right)$ in list $L_{2}$, then send $F_{1}\left(u_{i1}\right)$ back to A as the query response. Otherwise, C randomly picks $F_{1}\left(u_{i1}\right)\in {\left\{0,1\right\}}^{l_{1}}$, then updates the list $L_{2}=\left(L_{2},\left(u_{i1},F_{1}\left(u_{i1}\right)\right)\right)$, and finally outputs $F_{1}\left(u_{i1}\right)\in {\left\{0,1\right\}}^{l_{1}}$ as the response.$F_{2}$-query: the challenger C maintains a $F_{2}$ list $L_{3}=(F_{1}\left(u_{i1}\right),F_{2}\left(F_{1}\left(u_{i1}\right)\right)$, and set it empty in the beginning. When there is a $F_{1}$ query for $u_{i1}$ from adversary A, the challenger C firstly checks if it is in the $L_{4}$ list. If there is a corresponding pair $(F_{1}\left(u_{i1}\right),F_{2}\left(F_{1}\left(u_{i1}\right)\right)$, return $F_{2}\left(F_{1}\left(u_{i1}\right)\right)$, otherwise, challenger randomly chooses $F_{2}(F_{1}\left(u_{i1}\right)\in {\left\{0,1\right\}}^{l_{2}}$, then updates the list $L_{3}=(L_{3},\left(F_{1}\left(u_{i1}\right),F_{2}\left(F_{1}\left(u_{i1}\right)\right)\right)$, and finally outputs $F_{2}\left(F_{1}\left(u_{i1}\right)\right)\in {\left\{0,1\right\}}^{l_{2}}$ as the response.$H_{3}$-query: the challenger C maintains a $H_{3}$ list $L_{4}=\left(r_{i},u_{i2},C_{i}\right)$, and also sets the list as an empty set in the initial phase. When there is a query for $\left(r_{i},u_{i2}\right)$, the challenger C firstly checks if it is in the list. If it exists, then return the corresponding array $\left(r_{i},u_{i2},C_{i}\right)$ to A. Otherwise, C randomly selects vector $C_{i}\in {\left\{-1,0,1\right\}}^{N\times N}$, then updates the list $L_{4}=\left(L_{4},\left(r_{i},u_{i2},C_{i}\right)\right)$, and finally outputs $C_{i}$ as the response.KeyExtract-query: the challenger C maintains a KeyExtract list $L_{5}=(ID_{i},$$sk_{ID_{i}})$, and makes the list an empty set in the beginning. Now if the adversary A initiates a request for the private key associated with an identity $ID_{i}$, the challenger C checks if it is already in the $L_{5}$ list. If there exists the corresponding pair $\left(ID_{i},sk_{ID_{i}}\right)$, then the challenger C returns $sk_{ID}$. Otherwise C recovers the corresponding $\left(ID_{i},H_{1}\left(ID_{i}\right)\right)$ from the $L_{0}$ list, then C runs $\mathrm{GaussianSampler}(c,\sigma ,H_{1}\left(ID_{i}\right),0))$ to obtain $sk_{ID_{i}}=\left(s_{i_{1}},s_{i_{2}}\right)$, then updates the list $L_{5}=\left(L_{5},ID_{i},sk_{ID_{i}}\right)$.DelGen-query: the challenger C maintains a DelGen list $L_{6}=(y_{i_{1}},y_{i_{2}},u_{o_{2}},z_{i_{1}},$$z_{i_{2}})$ where warrant $W_{i}=u_{o_{1}}\|u_{o_{2}}$, When the adversary A issues a DelGen query for delegation of warrant $W_{i}$, the challenger $C_{i}$ searches it in $L_{6}$ list first, if there exist corresponding tuple $\left(y_{i_{1}},y_{i_{2}},u_{o_{2}},z_{i_{1}},z_{i_{2}}\right)$, return $z_{i_{1}},z_{i_{2}}$, otherwise, the adversary A executes $z_{i_{1}}=s_{o_{i_{1}}}C_{o}+y_{i_{1}},z_{i+1}=s_{o_{i_{2}}}C_{o}+y_{i_{2}}$ to obtain a valid delegation signature, then updates the list $L_{6}=\left(L_{6},y_{i_{1}},y_{i_{2}},u_{o_{2}},z_{i_{1}},z_{i_{2}}\right)$.Psign-query: the challenger C maintains a Psign list $L_{7}=\left(y_{i_{3}},y_{i_{4}},u_{p_{2}},z_{i_{3}},z_{i_{4}}\right)$ where message $U=u_{p_{1}}\|u_{p_{2}}$, when the adversary A issues a Psign query for the proxy signature of message *U*, the challenger C searches it in the $L_{7}$ list first, if there exists a corresponding tuple $\left(y_{i_{3}},y_{i_{4}},u_{p_{2}},z_{i_{3}},z_{i_{4}}\right)$, return $\left(z_{i_{3}},z_{i_{4}}\right)$. Otherwise, the adversary A executes $z_{i_{3}}=s_{p_{i_{1}}}C_{p}+y_{i_{3}},z_{i+1}=s_{p_{i_{2}}}C_{p}+y_{i_{4}}$ to obtain a valid proxy signature, then updates the list $L_{7}=\left(L_{7},y_{i_{3}},y_{i_{4}},u_{p_{2}},z_{i_{3}},z_{i_{4}}\right)$.Forgery After the interactions and queries, the adversary A outputs a valid forgery $\left(u_{o_{2}},u_{p_{2}},z_{i_{1}},z_{i_{2}},z_{i_{3}},z_{i_{4}}\right)$ with non-negligible probability on warrant *W*, message *U*, original signer identity $ID_{o}$ and proxy signer identity $ID_{p}$. We show that if A can do this forgery correctly then he is able to obtain a short non-zero solution of a SIS instance over NTRU lattice, i.e., the equation system $A_{h,q}\left(z_{1},z_{2}\right)=0$ mod *q* where $\|(z_{1},z_{2})\|<\beta$. The Queries phase can be executed again by A. According to the Forking lemma in [37] to generate another valid signature $\left(u_{o_{2}}^{*},u_{p_{2}}^{*},z_{i_{1}}^{*},z_{i_{2}}^{*},z_{i_{3}}^{*},z_{i_{4}}^{*}\right)$.
(6)$$H_{2}\left(hz_{i_{2}}+z_{i_{1}}-H_{1}\left({\mathrm{ID}}_{o}\right)C_{o}\right)=H_{2}\left(hz_{i_{2}}^{*}+z_{i_{1}}^{*}-H_{1}\left({\mathrm{ID}}_{o}\right){C_{o}}^{*}\right)$$
(7)$$H_{2}\left(hz_{i_{4}}+z_{i_{3}}-H_{1}\left({\mathrm{ID}}_{p}\right)C_{p}\right)=H_{2}\left(hz_{i_{4}}^{*}+z_{i_{3}}^{*}-H_{1}\left({\mathrm{ID}}_{p}\right)C_{p}^{*}\right)$$The following equation is true unless we can find a collision of the hash function $H_{2}$, which is hard in the random oracl model. So we can ensure their preimage is same.
$$hz_{i_{2}}+z_{i_{1}}-H_{1}\left({\mathrm{ID}}_{o}\right)C_{o}=hz_{i_{2}}^{*}+z_{i_{1}}^{*}-H_{1}\left({\mathrm{ID}}_{o}\right){C_{o}}^{*}$$
$$hz_{i_{4}}+z_{i_{3}}-H_{1}\left({\mathrm{ID}}_{p}\right)C_{p}=hz_{i_{4}}^{*}+z_{i_{3}}^{*}-H_{1}\left({\mathrm{ID}}_{p}\right)C_{p}^{*}$$Rearranging the two sides in the two equations, we obtain
$$h\left(z_{i_{2}}-z_{i_{2}}^{*}\right)+z_{i_{1}}-z_{i_{1}}^{*}+H_{1}\left({\mathrm{ID}}_{o}\right)\left(C_{o}^{*}-C_{o}\right)=0$$
$$h\left(z_{i_{4}}-z_{i_{4}}^{*}\right)+z_{i_{3}}-z_{i_{3}}^{*}+H_{1}\left({\mathrm{ID}}_{p}\right)\left(C_{p}^{*}-C_{p}\right)=0$$Since we have $s_{i}+s_{i+1}h=H_{1}\left({\mathrm{ID}}_{i}\right)$. We obtain
$$h\left(z_{i_{2}}-z_{i_{2}}^{*}\right)+z_{i_{1}}-z_{i_{1}}^{*}+\left(s_{1}+s_{2}h\right)\left(C_{o}^{*}-C_{o}\right)=0$$
$$h\left(z_{i_{4}}-z_{i_{4}}^{*}\right)+z_{i_{3}}-z_{i_{3}}^{*}+\left(s_{3}+s_{4}h\right)\left(C_{p}^{*}-C_{p}\right)=0$$Focusing on *h*, we have
$$h\left(z_{i_{2}}-z_{i_{2}}^{*}+s_{i_{2}}C_{o}^{*}-s_{i_{2}}C_{o}\right)+z_{i_{1}}-z_{i_{1}}^{*}+s_{i_{1}}C_{o}^{*}-s_{i_{1}}C_{o}=0$$
$$h\left(z_{i_{4}}-z_{i_{4}}^{*}+s_{i_{4}}C_{p}^{*}-s_{i_{4}}C_{p}\right)+z_{i_{3}}-z_{i_{3}}^{*}+s_{i_{3}}C_{p}^{*}-s_{i_{3}}C_{p}=0$$Then, we write the equations in matrix form, which are
$$\left(\begin{array}{c} h\\ 1 \end{array}\right)\left(\begin{array}{ccc} z_{i_{2}}-z_{i_{2}}^{*}+s_{i_{2}}C_{o}^{*}-s_{i_{2}}*C_{o} & \; & \end{array}z_{i_{1}}-z_{i_{1}}^{*}+s_{i_{1}}C_{o}^{*}-s_{i_{1}}*C_{o}\right)=0$$

$$\left(\begin{array}{c} h\\ 1 \end{array}\right)\left(\begin{array}{ccc} z_{i_{4}}-z_{i_{4}}^{*}+s_{i_{4}}C_{p}^{*}-s_{i_{4}}*C_{p} & \; & \end{array}z_{i_{3}}-z_{i_{3}}^{*}+s_{i_{3}}C_{p}^{*}-s_{i_{3}}*C_{p}\right)=0$$

As $\|\left(z_{i},z_{i}^{*}\right)\|\leq 2\sigma \sqrt{2N}$ and $\|\left(s_{i_{1}},s_{i_{1}}^{*}\right)\|\leq s\sqrt{2N}$ with overwhelming probability. We obtain
$$\|\left(z_{i_{2}}-z_{i_{2}}^{*}+s_{i_{2}}C_{o}^{*}-s_{i_{2}}*C_{o},z_{i_{1}}-z_{i_{1}}^{*}+s_{i_{1}}C_{o}^{*}-s_{i_{1}}*C_{o}\right)\|\leq \left(4\sigma +4s\lambda \right)\sqrt{2N}$$
$$\|\left(z_{i_{4}}-z_{i_{4}}^{*}+s_{i_{4}}C_{p}^{*}-s_{i_{4}}*C_{p},z_{i_{3}}-z_{i_{3}}^{*}+s_{i_{3}}C_{p}^{*}-s_{i_{3}}*C_{p}\right)\|\leq \left(4\sigma +4s\lambda \right)\sqrt{2N}$$Now if $(z_{i_{2}}-z_{i_{2}}^{*}+s_{i_{2}}C_{o}^{*}-s_{i_{2}}*C_{o},z_{i_{1}}-z_{i_{1}}^{*}+s_{i_{1}}C_{o}^{*}-s_{i_{1}}*C_{o})\neq 0$ and $(z_{i_{4}}-z_{i_{4}}^{*}+s_{i_{4}}C_{p}^{*}-s_{i_{4}}*C_{p},z_{i_{3}}-z_{i_{3}}^{*}+s_{i_{3}}C_{p}^{*}-s_{i_{3}}*C_{p})\neq 0$, it means that we can find an meaningful non-zero solution for a SIS instance in the NTRU lattice with overwhelming chance. Given Property 4 in [28] for Collision-Resistant preimage sampleable functions, the probability that algorithm C breaks the Short Integer Solution problem over the particular NTRU lattice is at least $(1-2^{\omega (\sqrt{\log N})})\varepsilon$.Therefore, assuming we are in random oracle model (ROM), if there is a PPT adversary A that can break the proposed IB-PSSMR over NTRU lattice with a non-negligible probability ϵ. Then we can use the algorithm A to construct a new PPT algorithm C to find a solution for the SIS problem in NTRU lattice. Additionally, which can be reduced to SVP problem over the NTRU lattice. So, assume the hardness of SVP problem, we claim our IB-PSSMR scheme is unforgeable. Given there is no known quantum algorithm for SVP, we can that claim our IB-PSSMR is also quantum resistant.Furthermore, it is not difficult to prove that our IB-PSSMR scheme is identifiability, strong undeniability, key dependence, and verifiability, for simplicity, we omit it here. □

## 6. Efficiency Analysis

At present, there are two kinds of security models for signature schemes, Random Oracle Model and Standard Model. Mostly, the more efficient lattice-based proxy signature schemes are those that proved secure in the random oracl model. Agrawal et al. [38] proposed a secure identity-based encryption scheme under the standard model, but their scheme is inefficient and can only encrypt one plaintext bit.

In this section, we will analyse some related proxy signature schemes and compare their metric with ours. We list the comparison of the signature length between our scheme and the related scheme under the same security parameter *N* setting, where m>5Nlogq, $\sigma =12\lambda m\sqrt{m}\omega \left(\sqrt{\log N}\right)$, *W* is the warrant, and *U* is the information to be signed.

From Table 1, the total length (signed message and signature) of scheme [39] is ∣W∣+∣U∣+4Nlog(12σ)+2N(logλ+1)=$|u_{o2}|+|u_{p2}|$+$2l_{2}$+4Nlog(12σ)+2N(logλ+1), the total length our message recovery signature scheme is $|u_{o2}|+|u_{p2}|+2|r|$+4Nlog(12σ)=$|u_{o2}|+|u_{p2}|+2l_{1}+2l_{2}$+4Nlog(12σ). Therefore, we make a proper reduction of 2N(logλ+1)$-2l_{1}$ in the communication overhead compared with [39] which is based on the NTRU lattice without message recovery.

Ducas et al. [40] proposed an efficient identity-based encryption (IBE) scheme based on NTRU lattice and a method to convert it into an identity-based signature (IBS) under the same framework. Compared with the scheme of [40], this paper adds the signature proxy authority and message recovery function. By constructing message recovery, in terms of transmission efficiency, our scheme can save communication bandwidth and only increase a small amount of computing resource consumption.

When we let security parameter N=512, we present the concrete instances of communication overhead reduction between our scheme and [39] in Table 2.

Furthermore, the energy consumption in transmission and computation is different. It is shown that a 32-bit computation requires less energy than a bit of transmission [29]. In our IB-PSSMR scheme, even if we make use of some more simpler computations, e.g., XOR and hash, in message recovery technology, we still obtain much less energy consumption than in the practical case [39].

Given the analysis above, we can conclude that the IB-PSSMR we refer to is more efficient than other lattice-based schemes in terms of communication and energy consumption.

## 7. Application of The IB-PSSMR

In this section, we discuss some application scenarios of our proposed IB-PSSMR scheme. Mostly, we will discuss its application in blockchain and Internet of Things.

For the proxy signature scheme, it is mainly about delegation authority. In the blockchain, the transfer of authority is often involved, such as transfer authority and certificate deposit authority [41]. In the cryptocurrency blockchain system, the private key of a wallet is usually held by a single node. However, in some cases, the currency of a wallet is publicly owned by an organization member, or it is necessary to give some proxy permissions to other nodes, which can exercise the same transfer permissions. At this time, the use of a proxy signature is needed. The frame diagram is shown in Figure 2. The wallet owning node will authorize the nodes within the organization with signature authority. The nodes that receive the legal proxy authorization can sign the transaction. After the signed transaction enters the transaction pool, it will be authenticated by the mining node to complete the confirmation of the transaction process. In the blockchain, to maintain the scalability of the blockchain, the block size of the blockchain will be strictly controlled. Therefore, the signature size of the transaction will also have an important impact on the performance of the blockchain. The IB-PSSMR scheme we proposed can compress the size of the signature well and can be used as an alternative signature algorithm for the post-quantum blockchain design.

In the Internet of Things environment, data authentication is of great significance [24,42,43]. Failure to perform integrity verification and authentication of data will lead to serious consequences. However, some edge nodes often have the problem of insufficient resource efficiency. Therefore, it is urgent to use a signature scheme that consumes fewer storage resources in the Internet of Things environment. Our proposed IB-PSSMR scheme can be used in future quantum computing environments in the Internet of Things scenario. For example, in the Internet of Things environment, an organization has many devices, one of which is the main device, and the other devices are also under the organization. At the same time, they share an identity. The proxy signature scheme can be used to authorize the affiliated devices. The traffic sent from the organization is the same identity. As shown in Figure 3, in the Internet of Things, the master device in the group can authorize the slave device by proxy. After the traffic sent by the slave device is signed by the proxy, it can be authenticated by other groups, and it can be attributed to the traffic of the same organization. Similarly, in this process, we need to control the size of the signature within a reasonable range, otherwise it will cause congestion to the traffic of the Internet of Things. The IB-PSSMR scheme can be used as an alternative to the post-quantum scheme in this Internet of Things environment to enhance data authentication.

## 8. Conclusions

Bandwidth is more precious than gold, especially in resource-constrained environments. In the era of quantum computing, it is necessary for us to construct an efficient proxy signature that is quantum safe. Because there are many post quantum schemes that use heavy computation and their signature size is not compact. The lattice- based architecture is the most attractive. In this paper, we construct an efficient identity-based proxy signature scheme with message recovery (IB-PSSMR) over the NTRU lattice under the standard Gentry–Peikert–Vaikuntanathan (GPV) framework [44]. In spite of the well-studied security proof, our scheme also benefits the excellent computation performance in NTRU lattice and can achieve the message recovery function in the sign phrase. We also give a formal security proof of our proposed scheme, and the efficiency analysis is compared with some related proxy signature construction. In the future, we will continue to improve the usability of our scheme and survey the concrete application scenario of our scheme.

## Figures and Tables

**Figure 1 entropy-25-00454-f001:**
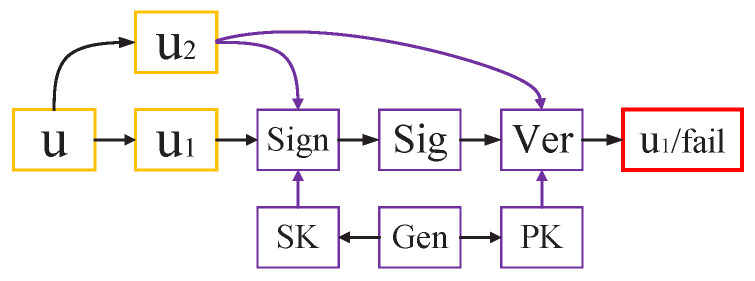
Signature with message recovery.

**Figure 2 entropy-25-00454-f002:**
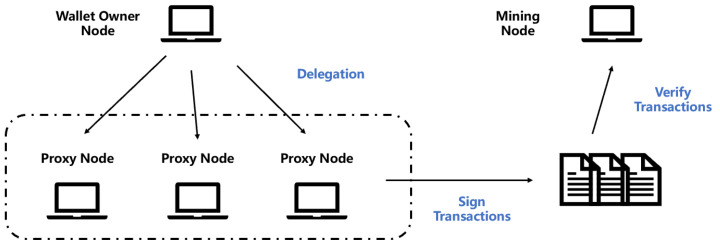
Proxy signature in blockchain.

**Figure 3 entropy-25-00454-f003:**
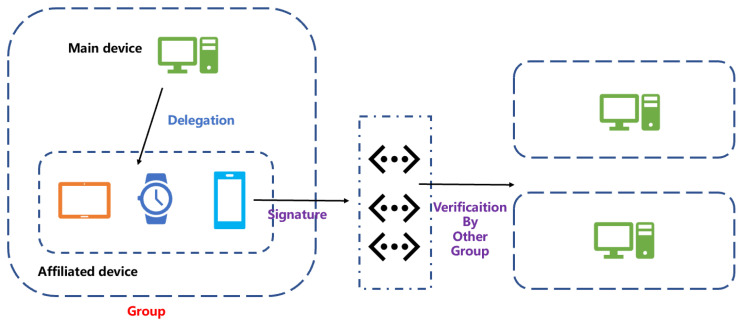
Proxy signature in IOT.

**Table 1 entropy-25-00454-t001:** Performance comparison among Refs. [39,40] and our scheme.

	Message Recovery	Delegation	Signature’s Size
[39]	No	Yes	$|u_{o2}|+|u_{p2}|$+$2l_{2}$+4Nlog(12σ)+2N(logλ+1)
[40]	No	No	N⌈logq⌉
Ours	Yes	Yes	$|u_{o2}|+|u_{p2}|+2l_{1}+2l_{2}$+4Nlog(12σ)

**Table 2 entropy-25-00454-t002:** Approximate measure of some concrete parameter instance.

Parameter Size (N, Instance, q, k, λ, $l_{1}$)	Communication Overhead Reduction (Bits)
(512, 1, 227, 80, 28, 100)	2305
(512, 2, 225, 512, 14, 100)	1997
(512, 3, 233, 512, 14, 200)	1777

## Data Availability

No new data were created or analyzed in this study. Data sharing is not applicable to this article.

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
