# Peer review of "Identity-Based Proxy Signature with Message Recovery over NTRU Lattice"

_entropy, 2023, doi:10.3390/e25030454_

Round 1
Reviewer 1 Report
In the manuscript, the authors present a new identity-based proxy signature scheme over NTRU lattice with message recovery for low resource constrained environments, which show that it is more efficient that the other existing identity-based proxy signature schemes in terms of the size of the signature and the cost of energy. The innovation point is relatively better. However, the whole expression should be improved with major revisions. Concretely, there are the following problems that should be addressed.
1.Some expressions are less rigorous involving some abbreviation expressions, symbol expressions, and syntax errors. For example, the expressions “We also discussed some application scenarios of IB-PSSMR in blockchain and IOT” is in Abstract section and “In 1993, …the DSA signature… ”, in which the abbreviation ‘IOT’ and ‘DSA’ should write the full name on the first time for improving the reading from many fields; The expression “such as difficult problems of (Elliptic Cureve)…” is in Introduction section, in which the item ‘Cureve’ has a spelling mistake; The expression “And Our scheme IB-PSSMR over…is consists of” is with grammatical mistake. The expression “Because there are many post quantum scheme…” in conclusion section, in which ‘scheme’ should be ‘schemes’, and so on. There are some other similar errors that are not mentioned. Please check the whole manuscript.
2. Under this threat, many scholars began to study post quantum cryptography to prevent many important cryptosystems from failing directly after the advent of quantum computers. It is better for appropriately citing some recent quantum signature references using quantum computing.
SKC-CCCO: an encryption algorithm for quantum group signature. Quantum Inf Process 21, 328 (2022).
A verifiable arbitrated quantum signature scheme based on controlled quantum teleportation[J]. Entropy, 2022, 24(1): 111.
Quantum multi-proxy blind signature based on cluster state. Quantum Inf Process 21, 104 (2022).
Arbitrated quantum signature protocol with boson sampling-based random unitary encryption[J]. Journal of Physics A: Mathematical and Theoretical, 2020, 53(13): 135301.
Arbitrated quantum signature scheme with quantum walk-based teleportation. Quantum Inf Process 18, 154 (2019).
3. In Table 1, more than one related reference are expected to make comparisons for showing the better performance.
4. In the manuscript, there are some Chinese style expressions and the authors should improve the corresponding expressions. For example, the expression “the expected number of times this process will output a signature is M” in subsection 2.4 has two predicate verb with ‘output’ and ‘is’. Please check the whole manuscript.
Author Response
Dear Reviewer:
We wish to take this opportunity to thank your constructive comments and valuable recommendations. We have carefully revised the manuscript based on the reviewers' suggestions. Please check. We really hope that the level of writing has been substantially improved. We will be happy to edit the text further, based on helpful comments from the reviewers.
Our responses to your comments are listed below:
Reviewer #1:
Thank you very much for your comments. We first thank you very sincerely for agreeing with our idea of paper. According with your advice, we checked our paper carefully and modified the mistakes of grammars sentence by sentence.
Reviewer #1, Major Comment #1: Some expressions are less rigorous involving some abbreviation expressions, symbol expressions, and syntax errors. For example, the expressions “We also discussed some application scenarios of IB-PSSMR in blockchain and IOT” is in Abstract section and “In 1993, …the DSA signature… ”, in which the abbreviation ‘IOT’ and ‘DSA’ should write the full name on the first time for improving the reading from many fields; The expression “such as difficult problems of (Elliptic Cureve)…” is in Introduction section, in which the item ‘Cureve’ has a spelling mistake; The expression “And Our scheme IB-PSSMR over…is consists of” is with grammatical mistake. The expression “Because there are many post quantum scheme…” in conclusion section, in which ‘scheme’ should be ‘schemes’, and so on. There are some other similar errors that are not mentioned. Please check the whole manuscript.
Author response: We thank the reviewer for pointing out these spelling errors and grammatical mistakes. We revised the manuscript and added the full name of abbreviations when they first appeared. We sincerely hope that now the new version of manuscript can give you a better reading experience. Please check.
Reviewer #1, Major Comment #2: Under this threat, many scholars began to study post quantum cryptography to prevent many important cryptosystems from failing directly after the advent of quantum computers. It is better for appropriately citing some recent quantum signature references using quantum computing.
SKC-CCCO: an encryption algorithm for quantum group signature. Quantum Inf Process 21, 328 (2022).
A verifiable arbitrated quantum signature scheme based on controlled quantum teleportation[J]. Entropy, 2022, 24(1): 111.
Quantum multi-proxy blind signature based on cluster state. Quantum Inf Process 21, 104 (2022).
Arbitrated quantum signature protocol with boson sampling-based random unitary encryption[J]. Journal of Physics A: Mathematical and Theoretical, 2020, 53(13): 135301.
Arbitrated quantum signature scheme with quantum walk-based teleportation. Quantum Inf Process 18, 154 (2019).
Author response: Thank you very much for your comments. We agree with your helpful suggestion. Hence, we have included the discussion of quantum computing in the introduction and cited these papers. Please check.
Reviewer #1, Major Comment #3: In Table 1, more than one related reference are expected to make comparisons for showing the better performance.
Author response: Thank you very much for your comment. We agree with your valuable suggestion and have added a new reference [39] in Table 1. We compared with [39]'s solution on message recovery and delegation support. Also, we added a discussion of Ref[37] in the efficiency analysis section. Please check.
Reviewer #1, Major Comment #4: In the manuscript, there are some Chinese style expressions and the authors should improve the corresponding expressions. For example, the expression “the expected number of times this process will output a signature is M” in subsection 2.4 has two predicate verb with ‘output’ and ‘is’. Please check the whole manuscript.
Author response: We apologize for the poor language of our manuscript. We modified this error and checked other grammatical problems in the article. We hope this will increase your reading experience. Please check.
Thank you for your careful review. We all really appreciate your efforts in reviewing our manuscript. We wish good health to you, your family, and community. Your careful review has helped to make our study clearer and more comprehensive.
Thanks very much for your attention to our paper.
Very sincerely yours,
Faguo Wu
Bo Zhou
Xiao Zhang
Reviewer 2 Report
Identity-based Proxy Signature with Message Recovery over NTRU Lattice
Authors: Faguo Wu, et al
The paper provides a new identity-based proxy signature scheme on NTRU lattice with message recovery which is more efficient than the other identity-based proxy signature in terms of the size of the signature and the cost of energy. Amongst many important areas such as identity-based proxy signature
The organization of the paper is sensible, and it allows the reader to get familiar with the concept of
lattice based cryptography that allows for extended functionality and is, at the same time, more efficient for basic primitives of public-key encryption and digital signature schemes.
My recommendation is based on the following issues that the authors may find interesting to consider:
(a) Major issue: I think the subject and the issues are interesting, however, there are many concerns that must be addressed, in particular the authors never discuss in detail their contribution, how their scheme is better than previous results-like identity-based encryption [Agrawal et al. 2010; Ducas et al. 2014- Efficient Identity-Based Encryption over NTRU Lattices. In Proceedings of the International Conference on The Theory and Application of Cryptology and Information Security. In its current form, it's not clear what this manuscript adds to the existing literature, I need more explanation on current NTRU cryptosystem.
Author Response
Dear Reviewer:
We wish to take this opportunity to thank your constructive comments and valuable recommendations. We have carefully revised the manuscript based on the reviewers' suggestions. Please check. We really hope that the level of writing has been substantially improved. We will be happy to edit the text further, based on helpful comments from the reviewers.
Our responses to your valuable comments are listed below:
We are grateful for your valuable and thoughtful comments. Before we respond the following questions, we first thank you very sincerely for your careful reading, helpful comments, and agreeing with our idea of paper. According to your advice, we have checked our paper carefully and modified the previous version of the manuscript. We really hope that the quality of the paper has been substantially improved and will be happy to edit the text further, based on helpful comments from you.
Reviewer, Comment #1: The paper provides a new identity-based proxy signature scheme on NTRU lattice with message recovery which is more efficient than the other identity-based proxy signature in terms of the size of the signature and the cost of energy. Amongst many important areas such as identity-based proxy signature
The organization of the paper is sensible, and it allows the reader to get familiar with the concept of
lattice based cryptography that allows for extended functionality and is, at the same time, more efficient for basic primitives of public-key encryption and digital signature schemes.
Author response: We are grateful to you for your effort reviewing our paper and positive feedback.
Reviewer, Comment #2: I think the subject and the issues are interesting, however, there are many concerns that must be addressed, in particular the authors never discuss in detail their contribution, how their scheme is better than previous results-like identity-based encryption [Agrawal et al. 2010; Ducas et al. 2014- Efficient Identity-Based Encryption over NTRU Lattices. In Proceedings of the International Conference on The Theory and Application of Cryptology and Information Security.
In its current form, it's not clear what this manuscript adds to the existing literature, I need more explanation on current NTRU cryptosystem.
Author response: We thank the reviewer for reading our paper carefully and giving the above comments. We have included the discussion of Agrawal2010 and Ducas2014 in the section of Efficiency Analysis. In fact, Agrawal et al. provided an Identity-Based Encryption (IBE) scheme, while our scheme is Identity-Based Signature (IBS). And the encryption scheme of Agrawal et al. can only encrypt one plaintext bit. The security analysis under the standard model provides theoretical value, but it is not very practical and inefficient. Our scheme is constructed based on the assumption of Random Oracle and can sign long messages. In the article "Efficient Identity-Based Encryption over NTRU Lattices", Ducas et al. proposed an IBE scheme based on NTRU lattice and the ability to convert it into an IBS scheme. Compared with the solution of Ducas2014, this paper adds the construction of proxy function and message recovery. Through message recovery, in terms of transmission efficiency, the solution of this paper can save communication bandwidth and only increase the consumption of a small amount of computing resources. Please check.
Thank you for your careful review. We all really appreciate your efforts in reviewing our manuscript. We wish good health to you, your family, and community. Your careful review has helped to make our study clearer and more comprehensive.
Thanks very much for your attention to our paper.
Very sincerely yours,
Faguo Wu
Bo Zhou
Xiao Zhang
Round 2
Reviewer 1 Report
Accept.
Reviewer 2 Report
I thank the authors for their reply. According to my questions, they made corresponding changes in the manuscript. I now recommend the manuscript for publication.